# Fast Hadamard-Encoded 7T Spectroscopic Imaging of Human Brain

**DOI:** 10.3390/tomography11010007

**Published:** 2025-01-13

**Authors:** Chan Hong Moon, Frank S. Lieberman, Hoby P. Hetherington, Jullie W. Pan

**Affiliations:** 1MR Research Center, Department of Radiology, University of Pittsburgh, 200 Lothrop Street, Pittsburgh, PA 15213, USA; 2Hillman Cancer Center, Department of Neurology, University of Pittsburgh, 5115 Centre Ave., Pittsburgh, PA 15232, USA; fsl1@pitt.edu; 3Resonance Research Inc., 31 Dunham Rd., Billerica, MA 01821, USA; hetheringtonh@icloud.com; 4NextGen Precision Health, Department of Radiology, University of Missouri Columbia, 1030 Hitt Street, Columbia, MO 65201, USA; jpv9m@umsystem.edu

**Keywords:** rosette trajectory, fast spectroscopic imaging, Hadamard slice encoding, 7T, tumor

## Abstract

**Background/Objectives**: The increased SNR available at 7T combined with fast readout trajectories enables accelerated spectroscopic imaging acquisitions for clinical applications. In this report, we evaluate the performance of a Hadamard slice encoding strategy with a 2D rosette trajectory for multi-slice fast spectroscopic imaging at 7T. **Methods**: Moderate-TE (~40 ms) spin echo and J-refocused polarization transfer sequences were acquired with simultaneous Hadamard multi-slice excitations and rosette in-plane encoding. The moderate spin echo sequence, which targets singlet compounds (i.e., N-acetyl aspartate, creatine, and choline), uses cascaded multi-slice RF excitation pulses to minimize the chemical shift dispersion error. The J-refocused sequence targets coupled spin systems (i.e., glutamate and myo-inositol) using simultaneous multi-slice excitation to maintain the same TE across all slices. A modified Hadamard slice encoding strategy was used to decrease the peak RF pulse amplitude of the simultaneous multi-slice excitation pulse for the J-refocused acquisition. **Results**: The accuracy of multi-slice and single-slice rosette spectroscopic imaging (RSI) is comparable to conventional Cartesian-encoded spectroscopic imaging (CSI). Spectral analyses for the J-refocused studies of glutamate and myo-inositol show that the Cramer Rao lower bounds are not significantly different between the fast RSI and conventional CSI studies. Linear regressions of creatine/N-acetyl aspartate and glutamate/N-acetyl aspartate with tissue gray matter content are consistent with literature values. **Conclusions**: With minimal gradient demands and fast acquisition times, the 2.2 min to 9 min for single- to four-slice RSI acquisitions are well tolerated by healthy subjects and tumor patients, and show results that are consistent with clinical outcomes.

## 1. Introduction

The primary challenges for MR spectroscopic imaging (MRSI) at 7T stem from amplitude-limited and inhomogeneous B_1_^+^ fields and a linear increase in B_0_ inhomogeneity (in Hz). Combined, these issues limit the quality of MRSI at 7T and likely contribute to the lack of an FDA-approved MRSI acquisition method on the commonly used 7T platforms. Thus, MRSI continues to be a focus for methodological development at 7T and 3T [1,2,3,4,5,6,7,8].

Our group has previously described MRSI at 7T using slice-based acquisitions without gradient-based outer volume suppression to minimize chemical shift dispersion error (CSDE) effects and an RF transceiver array to minimize B_1_^+^ inhomogeneity. Also, we have implemented improved shim strategies for enhanced B_0_ homogeneity and fast 2D spatial encoding using rosette trajectories (RSI) [9,10,11,12]. However, to extend these methods to multi-slice acquisitions with acceptable resolution and acquisition times (less than 10min), spatial accuracy and spectral quality must be maintained. For narrowly spaced contiguous slices, multi-slice coverage can be achieved using slab selection and conventional phase encoding across the slices. However, for low slice numbers, e.g., two, four, or eight slices, the slice profile is poorly defined due to the point spread function of the sampling. Further, the slab thickness increases the CSDE effects, creating artifacts in the edge slices. Originally described for its combinatorial properties, Bolinger and Leigh [13] and Goelman and Leigh [14] recognized the advantage of Hadamard encoding for the localization of a limited number of slices. In this context, Hadamard encoding can be considered to be a hybrid of (conventional individual) gradient-based slice selection and phase-encoded simultaneous multi-slice selection. Similar to conventional gradient-based slice selection, Hadamard encoding is beneficial since it largely retains the slice profile and CSDE of the chosen selective pulse. Similar to phase-encoded slices, Hadamard encoding is also beneficial, since each of the individual encoded slices accrues all of the available signal to noise ratio (SNR) according to the square root of the number of slices or acquisitions [15,16].

Thus, with Hadamard slice encoding, two to eight slices can be easily acquired. In this sequence, RF encoding can be implemented using either cascaded or simultaneous excitation pulses (i.e., multiplexed single slices) (Figure 1). Using cascaded pulses, the CSDE is minimized, although this results in each slice having a different TE. For singlet or non-J modulating resonances with transverse relaxation values of ~100–200 ms at 7T [17], the moderate increase in TE per slice (~7 ms) has a minimal effect on their overall amplitude for two or four slices (14 or 28 ms). In contrast, for J-modulating resonances and J-refocusing sequences where the result is dependent upon TE, the use of simultaneous multi-slice (SMS) excitation pulses is highly beneficial, as they eliminate the variation in TE. However, the required peak B_1_^+^ amplitude for SMS pulses is increased, resulting in an increase in CSDE due to limitations in achievable maximal B_1_^+^. In this work, we describe two different MRSI acquisitions targeting the singlet resonances N-acetyl aspartate (NAA), creatine (Cr), and choline (Cho) with a moderate-TE single-spin-echo sequence and glutamate (Glu) and myo-inositol (myo-Ins) using a J-refocused sequence [10]. Using a nominal voxel resolution of <0.75 cc and two temporal interleaves, two- and four-slice acquisitions were acquired in ~4.4 min to 8.8 min, depending on the desired SNR. For greater slice coverage for NAA, Cr, and Cho, eight-slice moderate-TE RSI required 17.6 min.

To evaluate the performance of the sequences, we acquired multi-slice Hadamard-encoded moderate-TE single-spin-echo RSI (four slices, 8.8 min) and single- (2.2 min) and multi-slice Hadamard-encoded J-refocused RSI (four slices, 8.8 min), along with a corresponding 14.5 min single-slice conventional 2D phase-encoded CSI study. Linear regression analysis of Cr/NAA with fraction gray matter (fGM) was used to evaluate the moderate-TE sequences in five healthy subjects. To quantitatively evaluate the J-refocused sequences, the rapid single-slice J-refocused RSI acquisition requiring 2.2 min (with a matched through-plane CSDE to conventional single-slice CSI) was directly compared to a conventional 14.5 min J-refocused CSI acquisition. For demonstration purposes, we also present data from a four-slice Hadamard-encoded J-refocused RSI sequence. Finally, to demonstrate the sensitivity of the sequences to pathology, we used the moderate-TE single-spin-echo RSI acquisition in two tumor patients, finding agreement of the MRSI data with the clinical scenario.

## 2. Materials and Methods

All studies were performed with a Siemens MAGNETOM 7T Step 2.3 system (Siemens Healthineers, Erlangen, Germany) equipped with an Avanto body gradient coil (maximum slew rate of 200 mT/m/ms, maximum gradient of 40 mT/m) and an eight-channel parallel transmit system. An 8 × 2 transceiver array (two rows in the head-to-foot direction, eight coils per row) was driven using eight 1→2 splitters, (Resonance Research Inc., Billerica, MA, USA). The Specific Absorption Rate (SAR) calculation for the transceiver coil was performed in global transmission mode. It should be noted, however, that even in circularly polarized (CP) mode, the transceiver gives a peak 10 g average SAR of 2.32 W/kg per 1 W of total input power [18], which is within the 10 W/kg local SAR for a six minute duration of the International Electrotechnical Commission (IEC) guideline [19]. As described previously [9], RF B_1_^+^ shimming was performed using two spatial B_1_^+^ distributions, targeting the whole brain (homogeneous distribution) and extracerebral regions (ring distribution). The ring B_1_^+^ distribution is used for suppression of the extracerebral tissues without gradient selection, while the homogeneous B_1_^+^ distribution is used for excitation, refocusing, and water suppression. Using this RF coil, a 750 Hz (17.6 uT) B_1_^+^ magnitude is readily achieved with less than 165 V per RF channel. Use of ring-based outer volume suppression eliminates the need for additional gradient-based in-plane localization, enabling a relatively short TR of 1.5 s to be used while remaining within the FDA guidelines for the SAR. Non-iterative B_0_ shimming was performed using a very high-order shim insert (VHOS, Resonance Research Inc., Billerica, MA, USA), as previously described [11], providing 1st–4th- and two 5th-order shims. Water suppression was achieved via a frequency-selective inversion recovery preparation module and optimized semi-selective refocusing pulses [9].

### 2.1. Multi-Slice Hadamard-Encoded Sequence: Cascaded and Simultaneous Excitation

Figure 1 shows the pulse sequences, including the slice-selective Hadamard encoding and 2D in-plane rosette readout components (blue oscillating waveform). Hadamard slice encoding is performed with RF encoding of the excitation pulse over a small number of slices, N_sl_ = 2, 4, or 8, and can be implemented with either a temporally cascaded (Figure 1A) or simultaneous RF pulse (Figure 1B). The temporally cascaded RF pulses were applied with the spin echo sequence, given its tolerance to variable TE. In comparison, as the J-refocused (double-echo) sequence is phase-sensitive (requiring that the entire signal is refocused at the end of the first spin echo [10]), simultaneous multi-slice excitation was used for all J-refocused studies.

As described [13,14,15,20], Hadamard localization uses RF-based phase encoding to generate spatial selectivity (rather than gradient-induced phase encoding) over a set of simultaneously acquired slices. The strategy applies 0° or 180° phase shifts to each of the slices according to the Hadamard encoding, thus allowing the extraction of individual slices through the application of the inverse Hadamard transform. Figure 1C shows the phase strategy for a four-slice Hadamard acquisition (M = 4). The four slices are denoted by slice A-B-C-D (signal S_A_ to S_D_), and to extract all four slices, four scans (S_1_ to S_4_) with each Hadamard encoding are needed. The left side of Figure 1C shows the scan-by-scan phase behavior for the excitation pulses for each slice. For four acquisitions, the Hadamard RF phases (in degrees) for the four slices are $\emptyset_{1}$ = (180,0,0,0), $\emptyset_{2}$ = (0,0,0,180), $\emptyset_{3}$ = (0,0,180,0), and $\emptyset_{4}$ = (180,0,180,180). The right side of Figure 1C shows the Hadamard reconstruction, i.e., S_slc_, which is calculated as(1)$$S_{slc}\left(z\right)=\sum_{m=1}^{M}S_{m}H_{slc,m}$$
where m is the scan index from [1, M], slc is the slice index from [A, D], and $H_{slc,m}$ is inverse Hadamard kernel (i.e., +1 or −1, as indicated) to calculate S_slc_ for the slc^th^ slice. For both sequences, a Shinnar–Le Roux optimized excitation pulse [21,22] with a maximum value of 750 Hz B_1_^+^ was used, resulting in a through-plane CSDE shift of 1 mm between NAA and Cr for a 7 mm slice. Because the full amplitude of B_1_^+^ is available per slice, the cascaded strategy is best to minimize the CSDE. This results in a range of echo times due to the added excitation pulses of approximately 7 ms per slice. To eliminate the phase roll that arises from this early acquisition start (for all but the last of the acquired slices), the calculated number of free induction decay (FID) points matching the added incremental echo times were deleted from the appropriate slices.

For the simultaneous excitation sequence, it is recognized that the RF phase strategy shown in Figure 1C as required by the Hadamard encoding can also be used to achieve simultaneous multi-slice SMS excitation without requiring excessive RF amplitudes (as would be conventionally needed). To increase the achievable selection bandwidth and minimize the CSDE for the 750 Hz maximum RF amplitude, we thus doubled the base RF pulse duration and applied RF phase encoding with summation over the four slice-selective pulses to form a single SMS pulse. As shown, this encoding scheme delivers at least one of the slice-selective pulses forming the single SMS pulse with the opposite phase to the other pulses on all acquisitions, thereby reducing the maximum required peak B_1_^+^ amplitude. The RF encoding between slices, however, can still be maintained through appropriate phase alternation of the individual RF selective excitations forming the single SMS pulse. For a four-slice sequence, this reduces the required B_1_^+^ peak from 2240 Hz to 650 Hz, achieving a per-slice bandwidth of 1100 Hz. With this B_1_^+^ amplitude reduction, a four-slice Hadamard-encoded J-refocused acquisition can be performed with a CSDE shift of ~2 mm between NAA and Cr for a set of four 7 mm slices. It should be cautioned, however, that with an increasing CSDE, the required polarization transfer between coupled spins required for the J-refocused sequence will deteriorate, resulting in a consistent but differential loss in signal between singlets and coupled resonances. Thus, for direct comparison of metabolite content and ratios (Glu, myo-Ins, and Glu/NAA) between the J-refocused RSI sequence and the Cartesian-encoded sequence, we used a 9 mm thick single-slice J-refocused RSI acquisition.

Finally, it should be noted that for either the cascaded or simultaneous sequence, because the refocusing pulses are not spatially selective, there is no refocusing variation across the Hadamard slices, allowing for simple (conventional) spatial encoding with the rosette trajectory.

### 2.2. Spatial Encoding with Rosette Trajectory and Curve Fitting

The rosette trajectory is an efficient strategy for 2D spectroscopic imaging (SI) with relatively low demands on gradient hardware, as has been published [23,24]. This current implementation traverses k-space using a quadrant circular trajectory that intersects the origin, k_xy_ = 0. Spectral encoding is achieved through repeated sampling through the circle, each circle being acquired with the desired spectral dwell time. Coverage of the $k_{xy}$ plane is achieved through multiple shots with angular rotation of the circle (size and number of shots $N_{sh}$ determined by pixel resolution, field of view, and the SNR). The analytic expression for the circular trajectory is(2)$$k_{xy}\left(t\right)=k_{max}\sin\left(2\pi ft\right)e^{-i2\pi ft}$$
where $k_{max}$ is the maximum spatial frequency band and 2f is the spectral bandwidth. It should be noted that, as the time derivative of the k trajectory in Equation (2), the magnitude gradient amplitude $\left|G\right|$ is constant (Equation (3))(3)$$\left|G\right|=\sqrt{G_{x}^{2}+G_{y}^{2}}={2\pi fk}_{max}$$
and thus the slew rate is also constant. Without any significant gradient ramp-up or ramp-down, this trajectory is highly efficient for sampling (there is minimal un-sampled time in the waveform), and given its constant gradient amplitude, eddy currents due to gradient switching are not significant. As is typically performed with spectral spatial sampling to achieve the needed spectral bandwidth at 7T and to minimize gradient demands, two temporal interleaves, $N_{it}$, are used; thus, this study used G_max_ = 5.5 mT/m and S_max_ = 40.2 mT/m/ms, with the number of acquired shots (a single shot is a repeatedly sampled single quadrant circle) being $N_{it}{\cdot N}_{sh}\cdot N_{sl}$, where $N_{sl}$ is the number of slices. For the control and patient scans, $N_{sh}$ = 44 was used; thus, a single-slice acquisition requires 2.2 min, and for a four-slice acquisition, the acquisition time is 8.8 min. In these studies, a 9 mm isotropic resolution (<0.65 cc nominal, 24 × 24 resolution over field of view, 216 × 216 mm^2^) was used, with a slice gap of 2 mm for the four-slice Hadamard encoding. The RSI data were reconstructed as previously described [23,24] using convolutional gridding with a Kaiser–Bessel kernel.

To compare the SNR of the moderate single-spin-echo Hadamard RSI (Figure 1A) vs. the Cartesian phase-encoded CSI, N_sh_ was chosen to match the single-slice CSI duration of 14.5 min, i.e., N_sh_ = 72. The same spectral processing was applied to the matched RSI and CSI acquisitions (7 mm thick, 9 × 9 mm^2^ in-plane nominal resolution) in two subjects. For the purposes of SNR comparison, 4 Hz Gaussian broadening and a 150 Hz convolution difference was applied. A linear combination model (LCM) was used for all spectral analyses (1.8 ppm to 4.0 ppm), with 15 compound basis functions (N-acetyl aspartate, N-acetyl aspartylglutamate, aspartate, lactate, creatine, phosphocreatine, γ-aminobutyric acid, glucose, glutamate, glutamine, glutathione, glycerophosphorylcholine, phosphorylcholine, myo-inositol, and taurine) calculated from GAMMA simulations incorporating the semi-selective refocusing profile [9,25,26].

Originally developed by Smith and Levante [26], GAMMA is a C++ library for density matrix simulation of magnetic resonance behavior with chemical structures. GAMMA is widely used for spectroscopic analysis and is available within the open-source Vespa package (Versatile Simulation, Pulses, and Analysis) for integrated use with curve fitting [27]. Spectral signal noise was directly measured from the standard deviation of 100 metabolite-free points at 6 ppm for all spectra. A Student’s *t*-test was used to compare the mean SNR, CRLB, and retained pixel counts between the RSI and CSI acquisitions, with *p* < 0.05 accepted for statistical significance.

The dependence of Cr/NAA on the tissue fraction of gray matter, fGM, was determined in five subjects. fGM is measured by using 2 magnetization-prepared rapid acquisition gradient echoes (MP2RAGE) images as(4)$$fGM={GM}_{voxel}/\left({GM}_{voxel}+{WM}_{voxel}\right)$$
where ${GM}_{voxel}$ and ${WM}_{voxel}$ are the voxels of GM and WM within an MRSI voxel. M2RAGE images were acquired at a 1 mm isotropic voxel resolution using TI_1_/TI_2_/TR = 0.9/1.6/5 s, with excitation angles of 5 and 9 degrees, respectively. The MP2RAGE images were segmented into GM, WM, and CSF using FreeSurfer (http://surfer.nmr.mgh.harvard.edu/). Using every other X pixel (with a Y pixel shift, giving 1/2 the available pixel numbers), linear regression analysis between fGM and Cr/NAA was applied, with significance taken at *p* < 0.05. As previously described [28], abnormal voxels were then identified using the tissue-defined fGM, slope (*m*), and intercept (*b*) calculated from five healthy subjects.

### 2.3. Human Subjects

All studies (control and patients) were performed with University of Pittsburgh Institutional Review Board oversight (IRB# STUDY20040095). There were five control volunteers (three females, age range 32 to 56; two males, age range 44 to 57, mean age 45 ± 11 yrs old). The clinical data were obtained from two brain tumor patients (47 and 69 yrs old), who were recruited from the clinical cases managed by the UPMC Tumor Board.

## 3. Results

### 3.1. Cascaded Excitation with Moderate-TE Acquisition

For the moderate-TE four-slice Hadamard acquisition, the cascaded RF approach was used to minimize the CSDE. The total acquisition time for the two data sets (Hadamard RSI vs. Cartesian CSI) was matched to 14.5 min. Figure 2A–C show the data from a healthy subject, with spectra from a matched conventional 2D phase-encoded CSI. For spectral analysis, voxels were included if the brain tissue volume (GM + WM) was >50%, linewidths were less than 0.11 ppm, and the CRLB for all major compounds (NAA, Cr, Cho, and myo-Ins) was <20%. To reject spectra with aberrant baselines, spectra were also excluded if the macromolecule (at 2.0 ppm)/Cr component was greater than 3.0. In this analysis, to eliminate the impact of the spatial overlap between adjacent pixels due to the point spread function and spatial filtering, every other pixel was included in the regression analysis (reducing the available pixels for analysis by two-fold). As the best metric of comparison between the Hadamard-encoded vs. conventionally encoded CSI, data from two subjects are shown in a plot of voxel-to-voxel integrated areas of NAA and Cr, demonstrating an R^2^ = 0.92 (Figure 2D). Using these inclusion criteria, (n = 140 voxels), the SNR of the 7 mm thick slice studies for NAA was 56.6 ± 30.6 vs. 63.9 ± 28.4, and the NAA CRLB was 2.2 ± 0.8 vs. 1.8 ± 0.8 for the CSI and RSI acquisitions, respectively, which were not significantly different.

In the five control volunteers, we evaluated the sensitivity of Cr/NAA to tissue GM content in the RSI data. Using co-registered T1 MP2RAGE structural images, the brain images were segmented and the GM content of each pixel calculated, including the point spread function weighting. Regressions of Cr/NAA with fGM were significant for all subjects and is shown for a volunteer in Figure 2E. Shown in Table 1 are the slopes (*m*) and intercepts (*b*) determined from regression of Cr/NAA against fGM for frontal, parietal, and all gray matter averaging across the five subjects. Appendix A shows the performance of this acquisition with eight slices.

### 3.2. J-Refocused RSI Acquisition

The J-refocused sequence improves the quantification of coupled resonances [10] by decreasing the effects of J-modulation for moderate TEs (up to 40 ms), while overlapping macromolecule signals are suppressed due to T2 relaxation. This is pertinent for metabolites such as Glu and myo-Ins, where spectral overlap dictates that linewidth, lipid contamination, and model accuracy are important for accurate spectral fitting [29,30]. Figure 3 shows 2.2 min spectra from a 9 mm thick single-slice J-refocused RSI acquisition in comparison to a 9 mm thick slice CSI acquisition for 14.5 min of equivalent volume resolution. Despite the significant difference in acquisition time, LCM curve fitting for these spectra shows no significant difference between the CRLB for Glu from either acquisition, with 4.8 ± 1.7% vs. 5.8 ± 2.1% for RSI and CSI, respectively (n = 2 subjects, ~130 pixels). For the myo-Ins CRLB, the RSI gave 6.3 ± 3.0%, the CSI 7.0 ± 3.6%, and there was no difference in the number of pixels meeting inclusion criteria. As expected, with similar filtering criteria as used above (including Glu and myo-Ins CRLB < 20%), a significant Glu/NAA regression with fGM is identified in all subjects (Figure 3D, Table 1).

Figure 4 shows the performance of the multi-slice J-refocused RSI. With the simultaneous (rather than cascaded) multi-slice excitation RF pulses, the CSDE shift in the slice direction is 2.2 mm between NAA and Cr. The SNR is consistent with the four-fold-longer acquisition time of 8.8 min used for the multi-slice study in comparison with the matched 2.2 min single-slice acquisition (Figure 4C).

### 3.3. Application in Tumor Patients

Figure 5 shows data from four-slice Hadamard-encoded moderate-TE spin echo RSI acquisitions in two tumor patients. Both patients had similar clinical histories, including a 5-to-10-year history of treatment and surveillance of a grade 2 oligodendroglioma. The data shown in Figure 5(A1–A3) are from a 47 yrs old tumor patient who was treated 5 years earlier with chemotherapy and external beam therapy. A mild non-enhancing lesion had been seen 2 weeks earlier on a 3T MRI, raising concerns about potential tumor recurrence in the region of the left frontal ventricle and basal ganglia. The 8.8 min scan single-spin-echo moderate-TE four-slice Hadamard study identified elevated Cho and decreased NAA in the area of the forceps minor, which gradually receded superiorly. This was consistent with clinical conclusions of low-grade tumor progression [31,32], with the patient proceeding to additional chemotherapy. Figure 5B shows data from a 69 yrs old patient whose tumor was initially treated 10 years earlier. Low-grade progression was noted 8 years later and again with the present study, with the patient again proceeding to additional chemotherapy.

## 4. Discussion

### 4.1. Feasibility of Multi-Slice Fast Spectroscopic Imaging

The high SNR at 7T is advantageous for spectroscopic imaging for acceleration and spectral quality. With work from several groups in targeting and developing rapid SI [1,33,34], the feasibility of multi-slice SI is of interest at 7T. At 3T, conventional B_0_ shimming over the entire brain for SI retains ~65% of all voxels with pixel loss typically occurring in the temporal and inferior frontal lobes [4,35]. This problem worsens at 7T and, thus, even with improved shim technology [11], the goal of maximizing spectral quality over regions of clinical interest argues for spatially targeted volumetric studies. We have implemented a multi-slice strategy from a rosette trajectory to achieve rapid acquisitions with better than 1cc resolution in one and four slices in 2.2 and 8.8 min, respectively. We find that the SNR of the multi-slice RSI is not significantly different from single-slice conventional CSI acquisitions of equal scan duration.

### 4.2. Spectral Quality and Curve Fitting

As a measure of spectral quality, the mean CRLB values between the 2.2 min RSI and 14.5 min CSI are similar (Figure 3) for high-concentration compounds such as Glu and myo-Ins. The lack of a strong dependence between the SNR and CRLB has been previously discussed for short-TE single-voxel spectra at 7T [36]. Multiple factors contribute to the CRLB of a resonance, including the need to account for lipid contamination and the macromolecule baseline signal in addition to the chosen spectral basis components, spectral overlap with other resonances, linewidth (LW), and the SNR. As discussed by Cavassila and experimentally verified at 3T for non-overlapping singlets at a moderate TE [5,30], the CRLB ∝ sqrt(LW)/SNR such that increasing the SNR gave proportionate decreases in the CRLB. However, with J-refocused and short-echo acquisitions, more complex spectral models are necessary. In the analysis of overlapping (and statistically correlated) resonances, increases in the SNR above a threshold may not necessarily generate additional decreases in the CRLB. Figure 6 shows this effect, comparing the CRLB, SNR, and linewidth for myo-Ins determined from 2.2 min and 6.6 min RSI (matched 9 mm isotropic, n = 2 subjects). While there is the expected SNR increase of 1.64 ± 0.51, the CRLB change is less, e.g., the mean ratio of CRLB_myo2.5_/CRLB_myo6.5_ is 1.21 ± 0.31 and CRLB_NAA2.5_/CRLB_NAA6.5_ is 1.21 ± 0.36. Instead, the experimental relationship between the CRLB and SNR for myo-Ins (Figure 6C) approaches a minimum of ~4.0% at a high SNR, similar to the 3% reported by Tkác at 7T for 8 cc single-voxel spectra from the occipital lobe [36]. A linear correlation between the CRLB and linewidth is significant, with R = 0.45, *p* < 0.001 (Figure 6D, identified despite the discrete coarse values reported by LCM for CRLB and linewidth). Thus, while earlier work has shown that for the major non-overlapping singlets the CRLB is proportional to sqrt(LW) [5], Figure 6D shows that the CRLB appears linearly proportional to LW for coupled overlapping resonances. This most likely reflects that the inverted Fisher information matrix (from which the CRLB values are calculated) for two overlapped peaks is a product correlation function between their linewidths and overlap (see Equation (21) from [20]). Overall, the ability to see these linewidth and SNR relationships is enabled by the fidelity of the fast J-refocused acquisition. With these gradient parameters and a moderate performance body gradient coil, the RSI is sufficiently accurate to encode the moderate-spin-echo and phase-sensitive polarization transfer J-refocused data.

### 4.3. Cascaded vs. Simultaneous Acquisition for Multiple Slices

There are several approaches for multi-slicing MRSI acquisitions for low slice numbers, e.g., Glover [37] and Souza [38]. In this report, we have used the Hadamard approach applied with either cascaded or simultaneous multi-slice excitation. The CSDE spatial shift is smallest with the cascaded method; however, the TE varies across the slices. Simultaneous RF slice encoding (i.e., one multi-band RF pulse exciting all slices) eliminates this effect but is limited by available peak B_1_^+^ amplitude for currently available 7T RF coils. We have demonstrated the utility of both these methods. A hybrid approach is also possible for the moderate-TE sequence, e.g., a 4 × 2 cascade of four simultaneous two-slice excitations would halve the extent of TE variation. A key issue is the peak B_1_^+^ amplitude reduction for the multi-band pulse, for which other methods are available, e.g., VERSE [39], or via modification of the target phase values for the individual slices [40,41]. We have used an alternative implementation of Hadamard RF phase encoding [13], which can be extended to higher orders, e.g., eight or sixteen slices. As a result, a further combination of these approaches may make simultaneous slice acquisitions using more slices feasible. With more widely spaced slices, further acceleration can also be achieved using simultaneous multi-slice methods given adequate receive coil complexity [42,43].

## 5. Conclusions

With acquisition times of <10 min, the current 7T MRSI methods give a spectral quality that is sufficient for the robust detection of GM-dependent increases in Cr/NAA and Glu/NAA. Notably, with a reasonable resolution (matrix size 24 × 24) and state-of-the-art gradient coils at 7T, the higher S_max_ and G_max_ achievable means that temporal interleaves are not strictly needed, making the minimum duration of single-slice and eight-slice acquisitions < 1.1 min and 8.8 min, respectively. However, eddy current corrections may be required due to the higher gradient amplitudes and slew rates. From an auditory tolerability perspective, these studies are very well accepted by subjects. Finally, as shown, the methods are sufficiently sensitive to identify the pertinent pathology in neuro-oncology patients. For future clinical applications, where relatively wide brain territories need to be sampled within 5 to 10 min, the developed simultaneous multi-slice fast MRSI acquisition with extended slice coverage (e.g., ~40 mm) is highly advantageous.

## Figures and Tables

**Figure 1 tomography-11-00007-f001:**
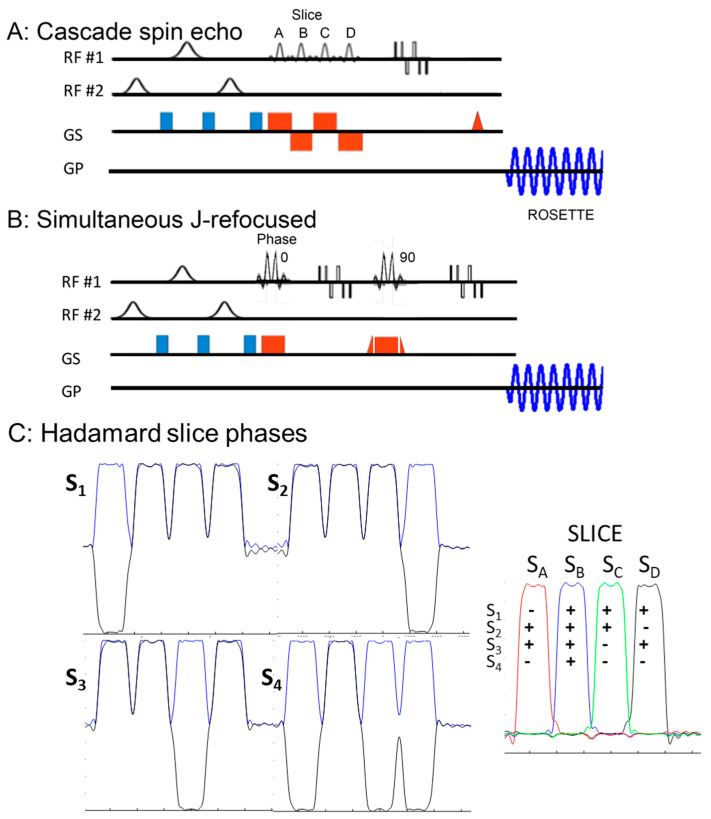
Pulse sequences for the (**A**) Hadamard slice encoding acquisition using a moderate-TE single-spin-echo with cascaded excitation pulses and the (**B**) J-refocused acquisition with simultaneous multi-slice excitation pulses. Four slices are shown in the cascaded sequence. For both sequences, two RF distributions are used, with the homogeneous distribution applied in the water suppression and spin echo components (RF #1) and the ring distribution (RF #2) applied for outer volume suppression of extracerebral signal. (**C**) The phases for the Hadamard-encoded excitation slice profiles of the sequence are shown in black for each scan (S_1_ through S_4_). The reconstruction scheme shows the summation of scans S_1_ to S_4_ needed to generate each of the slices (S_A_ through S_D_, red–blue–green–black). The pulses in the sequences are not shown to scale in magnitude or time.

**Figure 2 tomography-11-00007-f002:**
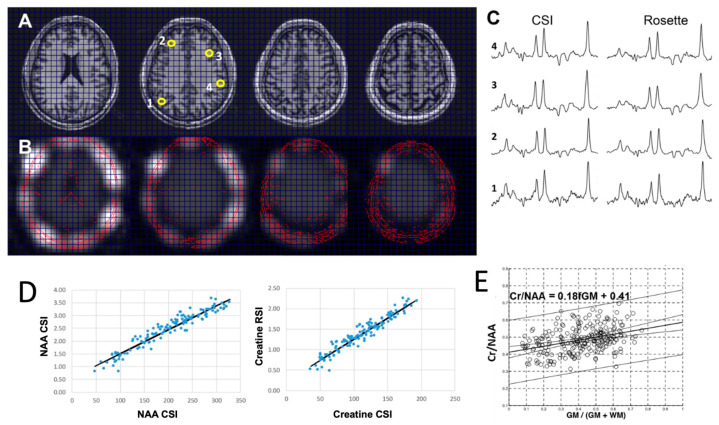
Comparison of data from matching CSI and four-slice Hadamard moderate-echo single-spin-echo RSI studies showing scout T1 anatomy (**A**), magnitude NAA images (**B**), spectra (**C**), and a voxel-to-voxel plot of NAA and Cr amplitudes (**D**). Four representative ROIs numbered 1 to 4 (yellow numbered circles in (**A**) match the spectra in (**C**). The regression analysis in (**D**) has R^2^ = 0.92 (*p* < 0.001). (**E**) Regression data for Cr/NAA vs. fraction GM (fGM) acquired from a control subject.

**Figure 3 tomography-11-00007-f003:**
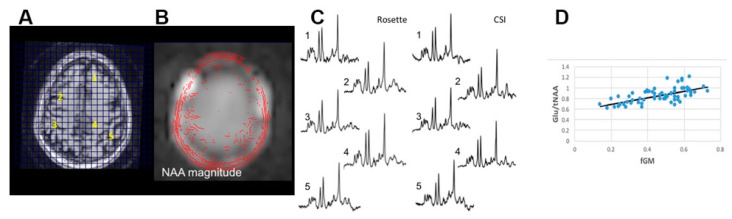
J-refocused single-slice spectroscopic imaging. (**A**) Scout T1, (**B**) magnitude NAA image, (**C**) comparison of matched spectra acquired with the RSI vs. the CSI from sampling points (yellow numbering, 1, 2, 3, 4, 5 in (**A**)), the corresponding numbered spectra in (**C**), and (**D**) regression of Glu/NAA with tissue fraction of GM (fGM) (every other pixel sampled), which has an R = 0.64, with a dependence of 0.57 + 0.61 × fGM (*p* < 0.001).

**Figure 4 tomography-11-00007-f004:**
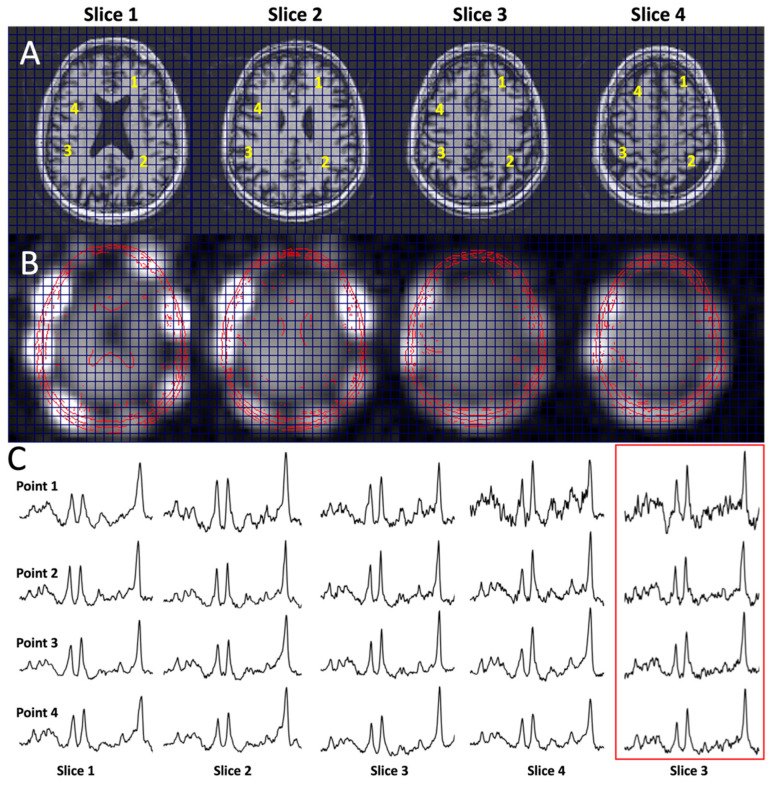
Multi-slice Hadamard J-refocused RSI. (**A**) Scout T1, (**B**) magnitude NAA images, and (**C**) sample spectra as indicated from four-slice J-refocused RSI using simultaneous Hadamard slice encoding; 7 mm slice thickness, 2 mm slice gap, ~9 min total duration, and TE 38 ms. Spectra at yellow numbered points (1, 2, 3, 4) in (**A**) for slice 1–4 are shown in (**C**); the red-boxed spectra shown in (**C**) are from the matched single-slice RSI for slice 3 for a 2.2 min duration.

**Figure 5 tomography-11-00007-f005:**
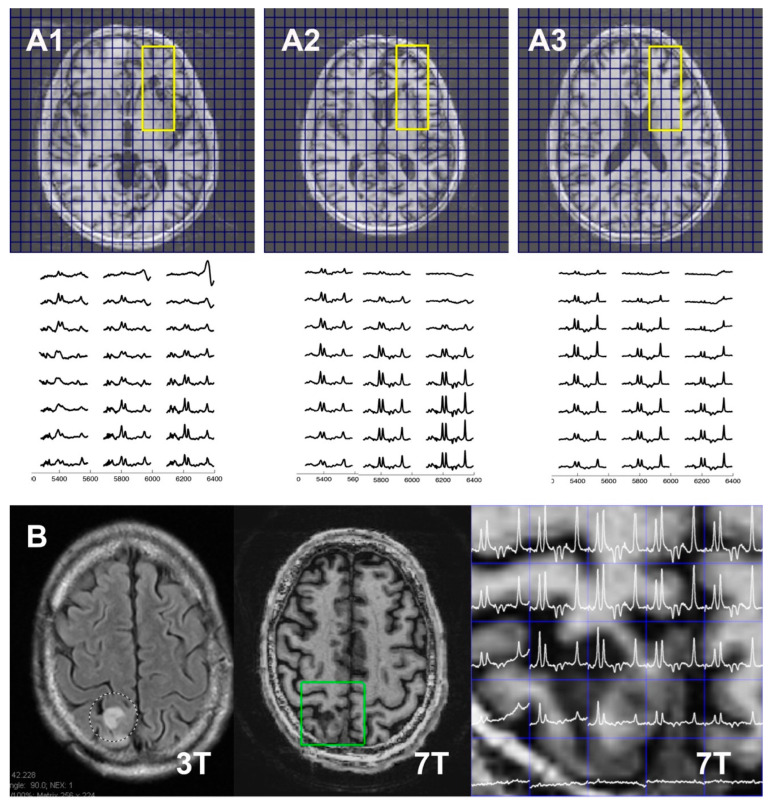
Moderate-TE spin echo cascaded Hadamard RSI acquisitions from two oligodendroglioma tumor patients, patient #1 (for slice 1 (**A1**), 2 (**A2**), and 3 (**A3**) in the top and middle panels); patient #2 ((**B**) in the bottom panel). In the (**A1**–**A3**) top panel, scout T1 anatomy is shown, while in the (**B**) two left panels, T2 FLAIR at 3T and T1 anatomy at 7T are shown. Spectra were from ROIs (yellow and green rectangle in (**A1**–**A3**) and (**B**), respectively). Both patients were clinically identified to have experienced tumor progression, consistent with spectroscopic imaging, although the 69 yo patient #2 shows a more aggressive worsening, with a Ch/NAA ratio of greater than 2 seen at the edge of the brain and lesion.

**Figure 6 tomography-11-00007-f006:**
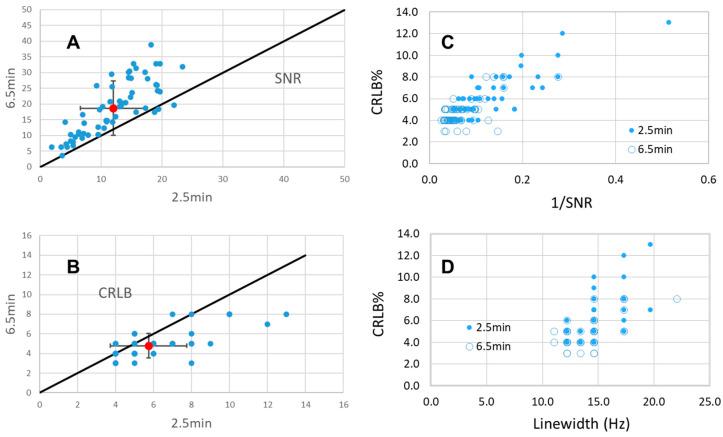
Spectral analyses of Inositol from 2.2 min and 6.6 min J-refocused RSI acquisitions at 9 mm isotropic resolution. (**A**,**B**) Comparison of 2.2 min vs. 6.6 min results for the SNR and CRLB. The red data point with error bars shows the mean and standard deviation for the plotted data; the black line shows the identity diagonal. Combining both the 2.2 min (filled circles) and 6.5 min (open circles) data, the relationship between 1/SNR and CRLB (**C**), linewidth (Hz), and the CRLB are shown (**D**). The plot in (**D**) appears to have fewer data points than in (**C**) due to the forced binned outputs of the LCM for both the CRLB and linewidth values (thus, each datapoint is representative of multiple points). The linear regressions calculated from all data points are significant for both 1/SNR and LW, with R^2^ = 0.67 and R^2^ = 0.38, *p* < 0.001 for both.

**Table 1 tomography-11-00007-t001:** Regression statistics for regression of Cr/NAA (acquired with the spin echo acquisition) and Glu/NAA (acquired with the J-refocused acquisition) with fraction GM (fGM) for five subjects. With the larger voxel size and higher slice used in the J-refocused acquisition, parcellation into frontal and parietal brain was not performed.

	Total GM			Frontal Region			Parietal Region		
	*b*	*m*	*Pixel #*	*b*	*m*	*Pixel #*	*b*	*m*	*Pixel #*
**Cr/NAA**	0.43 ± 0.08	0.20 ± 0.07	226 ± 13	0.39 ± 0.08	0.24 ± 0.06	48 ± 7	0.40 ± 0.14	0.32 ± 0.09	28 ± 6
**Glu/NAA**	0.49 ± 0.15	0.58 ± 0.06	67 ± 2	N/A					

The data represent mean ± standard deviation (n = 5) in slope (*m*), intercept (*b*), and pixel number (*Pixel #*).

## Data Availability

The processed data supporting the conclusions of this article will be made available by the authors on request.

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
