# Peer review of "Fast Hadamard-Encoded 7T Spectroscopic Imaging of Human Brain"

_tomography, 2025, doi:10.3390/tomography11010007_

Round 1
Reviewer 1 Report
Comments and Suggestions for Authors
In the article “Fast Hadamard encoded 7T spectroscopic imaging of human brain”, the authors evaluate the performance of a Hadamard slice encoding strategy with a 2D rosette trajectory for multi-slice fast spectroscopic imaging at 7T. The manuscript is a continuation of the authors’ research and is important in the area of knowledge discussed. For the article to proceed further it requires suggested corrections and clarifications:
1. The most recent literature cited by the authors is from 2018. Please update the literature.
2. I suggest that Figures and Tables should be placed in the section of the manuscript where the authors refer to/discuss them. This will make it easier for the potential reader of the data to follow.
3. All abbreviations used in the text of the article e.g. SNR, LCM should be explained.
4. Line 141: the authors write ‘...calculated from GAMMA simulations...’. Please discuss/explain.
5. Line 144, 146. Please explain the symbols used in the formulae. I suggest numbering all formulae cited in the paper.
6. In the section ‘Materials and methods’, please describe the methods of statistical analysis used.
7. Figure 2: In the title, explain the meaning of the numbers 1,2,3,4 in the graph.
8. Figure 3: In the title, please explain the meaning of the numbers 1,2,3,4,5 in the figure.
9. Figure 4: In the title, explain the meaning of the numbers 1,2,3,4 in the graph.
10. Please improve the quality of Table 1. I suggest replacing „” with ”±’.The table gives the impression of being incomplete. Please reword.
11. The authors write: Line 192 ‘...In five control volunteers (3 female, 2 males, mean age 45±11 yrs.)...’. Line 225 ‘...two tumour patients...’.
Study groups/control patients should be discussed in detail in the ‘Materials and Methods’ section.
Please tabulate the baseline characteristics of male and female subjects.
12. Line 226. The authors write:‘...The data shown in Fig. 5A are from a 47yo tumor patient who was treated 5 years earlier with chemotherapy and external beam therapy...’. Line 234 ‘...Fig. 5B shows data from a 69yo patient whose tumor was initially treated 10 years earlier...’. Do the authors have the patients' approval to publish their results?
13. In the conclusion, please describe the prospects for further research.
14. Line 383. Please provide the Bioethics Committee approval number and date of approval!
Author Response
We thank the editorial team and both reviewers for their thoughtful comments. The reviewer has provided valuable feedback that is very helpful in improving the potential contribution of this work. Responses to Reviewers’ comments are answered below in blue font. The red font text indicates the requested changes. Comment balloons have been added to the Revised Text, itemized according to the enumeration of the Reviewer’s comments (R1.1. and R2.1. etc.).
Review Report Form #1
Comments and Suggestions for Authors:
In the article “Fast Hadamard encoded 7T spectroscopic imaging of human brain”, the authors evaluate the performance of a Hadamard slice encoding strategy with a 2D rosette trajectory for multi-slice fast spectroscopic imaging at 7T. The manuscript is a continuation of the authors’ research and is important in the area of knowledge discussed.
For the article to proceed further it requires suggested corrections and clarifications:
R1.1. The most recent literature cited by the authors is from 2018. Please update the literature.
We agree with the reviewer’s point. Three recent papers related to MRSI studies are added in revised manuscript References.
Page 1, Line 39
Thus, MRSI continues to be a focus for methodological development at 7T and 3T (1-8).
Page 13, Line 478 - 484
- Bogner W, Otazo R, Henning A. Accelerated MR spectroscopic imaging-a review of current and emerging techniques. NMR Biomed 2021;34(5):e4314. doi: 10.1002/nbm.4314
- Hingerl L, Strasser B, Moser P, Hangel G, Motyka S, Heckova E, Gruber S, Trattnig S, Bogner W. Clinical High-Resolution 3D-MR Spectroscopic Imaging of the Human Brain at 7 T. Investigative Radiology 2020;55(4).
- Maudsley AA, Andronesi OC, Barker PB, Bizzi A, Bogner W, Henning A, Nelson SJ, Posse S, Shungu DC, Soher BJ. Advanced magnetic resonance spectroscopic neuroimaging: Experts' consensus recommendations. NMR in Biomedicine 2021;34(5):e4309. doi: https://doi.org/10.1002/nbm.4309
R1.2. I suggest that Figures and Tables should be placed in the section of the manuscript where the authors refer to/discuss them. This will make it easier for the potential reader of the data to follow.
Thanks for your suggestion to make the manuscript more readable. The figures and table have been redistributed to the corresponding sections in main text.
R1.3. All abbreviations used in the text of the article e.g. SNR, LCM should be explained.
The abbreviated words have been defined.
Signal to noise ratio (SNR), linear combination model (LCM), specific absorption rate (SAR), etc.
R1.4. Line 141: the authors write ‘...calculated from GAMMA simulations...’. Please discuss/explain.
The background on the GAMMA simulation (Smith and Levante 1994) is now summarized more completely including reference to Brian Soher’s VESPA package that provides access to the original GAMMA C++ library.
Page 5, Line 222 – 226
Originally developed by Smith and Levante (26), GAMMA is a C++ library for density matrix simulation of the magnetic resonance behavior with chemical structures. GAMMA is widely used for spectroscopic analysis and is available within the open-source Vespa package (Versatile Simulation, Pulses, and Analysis) for integrated use with curve fitting (27).
Page 14, Line 524 - 525
- Soher BJ, Semanchuk P, Todd D, Ji X, Deelchand D, Joers J, Oz G, Young K. Vespa: Integrated applications for RF pulse design, spectral simulation and MRS data analysis. Magn Reson Med 2023;90(3):823-838. doi: 10.1002/mrm.29686
R1.5. Line 144, 146. Please explain the symbols used in the formulae. I suggest numbering all formulae cited in the paper.
Fraction GM, fGM has been rewritten and the Equations are numbered.
Page 5 - 6, Line 230 – 234
The dependence of Cr/NAA on tissue fraction of gray matter, fGM was determined in five subjects. fGM is measured by using magnetization prepared 2 rapid acquisition gradient echoes (MP2RAGE) images as
fGM = GMvoxel/(GMvoxel + WMvoxel) [4]
where GMvoxel and WMvoxel is the voxels of GM and WM within a MRSI voxel.
R1.6. In the section ‘Materials and methods’, please describe the methods of statistical analysis used.
We have now stated the use of a Student’s t-test for comparison of SNR, CRLB and pixel counts in the Methods. There is a brief discussion of the statistics used to analyze the control datasets and the detection of abnormal voxels; this is based on previous work that we and others have published.
Page 5, Line 227 - 229
A Student’s t-test was used to compare mean SNR, CRLB and retained pixel counts between the RSI and CSI acquisitions, with p<0.05 accepted for statistical significance.
Page 6, Line 238 - 241
Using every other X pixel (with a Y pixel shift, giving ½ the available pixel numbers), linear regression analysis between fGM and Cr/NAA was applied with significance taken with p<0.05. As previously described (28), abnormal voxels were then identified using the tissue defined fGM, slope (m) and intercept (b) calculated from five healthy subjects.
R1.7. Figure 2: In the title, explain the meaning of the numbers 1,2,3,4 in the graph.
Figure 2 caption is modified including four ROIs numbering.
Page 7, Line 276 - 280
Figure 2. Comparison of data from matching CSI and four-slice Hadamard moderate echo single-spin-echo RSI studies showing scout T1 anatomy (A), magnitude NAA images (B), spectra (C) and a voxel-to-voxel plot of NAA and Cr amplitudes (D). Four representative ROIs numbered 1 to 4 (yellow numbered circles in A) match the spectra in C. The regression analysis in D has R2 = 0.92 (p < 1e-3). (E) Regression data for Cr/NAA vs. fraction GM (fGM) acquired from a control subject.
R1.8. Figure 3: In the title, please explain the meaning of the numbers 1,2,3,4,5 in the figure.
Figure 3 caption is modified including numbering of five sampling points.
Page 7, Line 299 - 303
Figure 3. J-refocused single-slice spectroscopic imaging. (A) Scout T1, (B) magnitude NAA image, (C) comparison of matched spectra acquired with the RSI vs. the CSI from sampling points (yellow numbering, 1,2,3,4,5 in A, the corresponding numbered spectra in C), and (D) regression of Glu/NAA with tissue fraction of GM (fGM) (every other pixel sampled) has a R = 0.64, with a dependence of 0.57 + 0.61´fGM (p < 1e-7).
R1.9. Figure 4: In the title, explain the meaning of the numbers 1,2,3,4 in the graph.
Figure 4 caption and panel naming is modified including numbering of five sampling points.
Page 8, Line 317 – 321
Figure 4. Multi-slice Hadamard J-refocused RSI. (A) scout T1, (B) magnitude NAA images, and (C) sample spectra as indicated from a four-slice J-refocused RSI using simultaneous Hadamard slice encoding; 7mm slice thickness, 2mm slice gap, ~9min total duration, and TE 38ms. Spectra at yellow numbered points (1,2,3,4) in A for slice 1 - 4 are shown in C; the red-boxed spectra shown in C are from the matched single-slice RSI for slice 3 at 2.2min duration.
R1.10. Please improve the quality of Table 1. I suggest replacing „” with ”±’.The table gives the impression of being incomplete. Please reword.
Yes, this has been fixed.
Page 8, Line 304 – 308
Table 1 Regression statistics for regression of Cr/NAA (acquired with the spin echo acquisition) and Glu/NAA (acquired with the J-refocused acquisition) with fraction GM (fGM) for five subjects. With the larger voxel size and higher slice used in the J-refocused acquisition, parcellation into frontal and parietal brain was not performed.
|
Total GM |
Frontal Region |
Parietal Region |
||||||
b |
m |
Pixel # |
b |
m |
Pixel # |
b |
m |
Pixel # |
|
Cr/NAA |
0.43 ± 0.08 |
0.20 ± 0.07 |
226 ± 13 |
0.39 ± 0.08 |
0.24 ± 0.06 |
48 ± 7 |
0.40 ± 0.14 |
0.32 ± 0.09 |
28 ± 6 |
Glu/NAA |
0.49 ± 0.15 |
0.58 ± 0.06 |
67 ± 2 |
N/A |
The data represents mean ± standard deviation (n=5) in slope (m), intercept (b) and pixel number (Pixel #).
R1.11. The authors write: Line 192 ‘...In five control volunteers (3 female, 2 males, mean age 45±11 yrs.)...’. Line 225 ‘...two tumor patients...’.
Study groups/control patients should be discussed in detail in the ‘Materials and Methods’ section.
Please tabulate the baseline characteristics of male and female subjects.
We have included the Human Subjects paragraph in the Materials and Methods.
Page 6, Line 243 - 248
All studies (control and patients) were performed with the University of Pittsburgh Institutional Review Board oversight (IRB# STUDY20040095). Five control volunteers (3 females age range 32 to 56; 2 males age range 44 to 57, mean age 45±11 yrs old). The clinical data were obtained from two brain tumor patients (47 and 69 yrs old) who were recruited from the clinical cases managed by the UPMC Tumor Board.
R1.12. Line 226. The authors write:‘...The data shown in Fig. 5A are from a 47yo tumor patient who was treated 5 years earlier with chemotherapy and external beam therapy...’. Line 234 ‘...Fig. 5B shows data from a 69yo patient whose tumor was initially treated 10 years earlier...’. Do the authors have the patients' approval to publish their results?
Yes, these patients were recruited from the UPMC Tumor Board as described in the Human Subjects paragraph (Materials and Methods).
Page 6, Line 246 - 248
The clinical data were obtained from two brain tumor patients (47 and 69 yrs old) who were recruited from the clinical cases managed by the UPMC Tumor Board.
R1.13. In the conclusion, please describe the prospects for further research.
Agree, we have stated the clinical advantage of the fast MRSI for clinical applications in the Conclusions.
Page 12, Line 430 - 430
For future clinical applications, where relatively wide brain territories need to be sampled within 5 to 10 min, the developed simultaneous multi-slice fast MRSI acquisition with extended slice coverage (e.g., ~40mm) is highly advantageous.
R1.14. Line 383. Please provide the Bioethics Committee approval number and date of approval!
Yes, this is now included in the Methods.
Page 6, Line 244 - 245
All studies (control and patients) were performed with the University of Pittsburgh Institutional Review Board oversight (IRB# STUDY20040095).
Page 13, Line 453 - 455
The study was conducted in accordance with the Declaration of Helsinki, and approved by the Institutional Review Board of the University of Pittsburgh (IRB# STUDY20040095, approved April 24, 2020).
Reviewer 2 Report
Comments and Suggestions for Authors
The paper proposes a new method of using Hadamard encoding to 7T spectroscopic imaging aimed at making human brain metabolites analysis both faster and of higher quality. However, there are still some concerns:
1. The authors are encouraged to elaborate on the mathematical underpinnings and physical principles behind the Hadamard encoding technique, including detailed derivations that explain its operational mechanisms and the reasons behind its experimental performance.
2 The authors should consider providing more in-depth details regarding the set of Hadamard transformations as well as the rosette trajectory sampling method.
3. Given the high field strength of 7T, the manuscript had better include a discussion on safety consideration, such as specific absorption rate and potential heating effects
4. The authors had better include objective metrics, such as contrast-to-noise ratio, spectral fidelity measures,essential in the evaluating the quality of images produced by their method..
5.The author had better conduct a comprehensive error analysis.
Author Response
We thank the editorial team and both reviewers for their thoughtful comments. The reviewer has provided valuable feedback that is very helpful in improving the potential contribution of this work. Responses to Reviewers’ comments are answered below in blue font. The red font text indicates the requested changes. Comment balloons have been added to the Revised Text, itemized according to the enumeration of the Reviewer’s comments (R1.1. and R2.1. etc.).
Review Report Form #2
Comments and Suggestions for Authors
The paper proposes a new method of using Hadamard encoding to 7T spectroscopic imaging aimed at making human brain metabolites analysis both faster and of higher quality. However, there are still some concerns:
R2.1. The authors are encouraged to elaborate on the mathematical underpinnings and physical principles behind the Hadamard encoding technique, including detailed derivations that explain its operational mechanisms and the reasons behind its experimental performance.
We have added more details of applied Hadamard encoding, including its original conceptualization from Bolinger and Leigh 1988 and Goelman and Leigh 1991. Given its lengthy history in the MR literature since then, we summarize the key principles that underlie the Hadamard strategy in the Introduction and in the Methods.
Page 2, Line 51 - 53
Originally described for its combinatorial properties, Bolinger and Leigh (13) and Goelman and Leigh (14) recognized the advantage of Hadamard encoding for localization of a limited number of slices. In this context, Hadamard encoding can be considered to be a hybrid of (conventional individual) gradient based slice selection and phase encoded simultaneous multi-slice selection. Similar to conventional gradient based slice selection, Hadamard encoding is beneficial since it largely retains the slice profile and CSDE of the chosen selective pulse. Similar to phase encoded slices, Hadamard encoding is also beneficial since each of the individual encoded slices accrues all of the available signal to noise ratio, SNR, according to the square root of the number of slices or acquisitions (15, 16).
Page 3, Line 124 - 141
As described (13-15, 20), Hadamard localization uses RF-based phase encoding to generate spatial selectivity (rather than gradient-induced phase encoding) over a set of simultaneously acquired slices. The strategy applies 0° or 180° phase shifts to each of the slices according to the Hadamard encoding, thus allowing the extraction of individual slices through application of the inverse Hadamard transform. Fig. 1C shows the phase strategy for a 4 slice-Hadamard acquisition (M = 4). The four slices are denoted by A-B-C-D (signal SA to SD), and to extract all 4 slices, four scans (S1 to S4) with each Hadamard encoding are needed. The left side of Fig. 1C shows the scan-by-scan phase behavior for the excitation pulses for each slice. For 4 acquisitions, the Hadamard RF phases (in degrees) for the 4 slices are =[180,0,0,0], =[0,0,0,180], =[0,0,180,0], and =[180,0,180,180]. The right side of Fig. 1C shows the Hadamard reconstruction, i.e., Sslc is calculated as
Sslc(z) = Σm=1,M {Sm*Hslc,m} [1]
where m is the scan index from [0, M-1], slc is the slice index from [A, D], and Hslc,m is inverse Hadamard kernel (i.e., +1 or -1, as indicated) to calculate Sslc for slcth slice. For both sequences, a Shinnar Leroux optimized excitation pulse (21, 22) with a maximum value of 750Hz B1 was used, resulting in a through-plane CSDE shift of 1mm between NAA and Cr for a 7mm slice. Because the full amplitude of B1+ is available per slice, the cascaded strategy is best to minimize the CSDE.
Page 13, Line 493 - 510
- Bolinger L, Leigh JS. Hadamard spectroscopic imaging (HSI) for multivolume localization. Journal of Magnetic Resonance (1969) 1988;80(1):162-167. doi: https://doi.org/10.1016/0022-2364(88)90070-4
- Goelman G, Leigh JS. B1-insensitive Hadamard spectroscopic imaging technique. Journal of Magnetic Resonance (1969) 1991;91(1):93-101. doi: https://doi.org/10.1016/0022-2364(91)90411-L
- Goelman G, Liu S, Gonen O. Reducing voxel bleed in Hadamard-encoded MRI and MRS. Magn Reson Med 2006;55(6):1460-1465. doi: 10.1002/mrm.20903
R2.2. The authors should consider providing more in-depth details regarding the set of Hadamard transformations as well as the rosette trajectory sampling method.
Please see above and in the text for the several changes describing the Hadamard technique.
We also now provide substantially more background on the non-Cartesian rosette trajectory. This method has also had reasonable published history in the MR literature, by Noll and Schirda. To maintain clarity in the paper but not re-publish this work, we have summarized the key principles that underlie its use in the Methods.
Page 4, Line 173 - 189
The rosette trajectory is an efficient strategy for 2D spectroscopic imaging with relatively low demands on gradient hardware as has been published(23, 24). This current implementation traverses k-space using a single circular trajectory that intersects the origin, kxy=0. Spectral encoding is achieved through the repeated sampling through the circle, each circle acquired with the desired spectral dwell time. Coverage of the plane is achieved through multiple shots with angular rotation of the circle (size and number of shots determined by pixel resolution, field of view, and SNR). The analytic expression for the circular trajectory is
kxy(t) = kmax*sin(2*pi*f*t)exp(-i2*pi*f*t) [2]
where is maximum spatial-frequency band and is the spectral bandwidth. It should be noted that as the time derivative of k trajectory in Eq.2, the magnitude gradient amplitude |G| is constant (Eq.2)
|G| = sqrt(Gx^2 +Gy^2) = 2*pi*f*kmax [3]
and thus the slew rate is also constant. Without any significant gradient ramp-up or ramp-down, this trajectory is highly efficient for sampling (there is minimal un-sampled time in the waveform), and given its constant gradient amplitude, eddy currents due to gradient switching are not significant. As typically done with spectral spatial sampling to achieve the needed spectral bandwidth at 7T and to minimize gradient demands, two temporal interleaves, are used; thus this study used Gmax=5.5 mT/m, Smax=40.2 mT/m/ms, with the number of acquired shots (a single shot is a repeatedly sampled single circle) being , where is the number of slices. For the control and patient scans, =44 was used; thus a single slice acquisition requires 2.2min; for the four-slice acquisition, the acquisition time is 8.8min. In these studies, 9mm isotropic (<0.65cc nominal, 24×24 resolution over field of view, 216×216mm2) was used with a slice gap of 2mm for the four-slice Hadamard encoding. The RSI data were reconstructed as previously described (23, 24) using convolutional gridding with a Kaiser-Bessel kernel.
Page 14, Line 516 - 519
- Noll DC. Multishot rosette trajectories for spectrally selective MR imaging. IEEE Trans Med Imaging 1997;16(4):372-377. doi: 10.1109/42.611345
- Schirda CV, Tanase C, Boada FE. Rosette spectroscopic imaging: optimal parameters for alias-free, high sensitivity spectroscopic imaging. J Magn Reson Imaging 2009;29(6):1375-1385. doi: 10.1002/jmri.21760
R2.3. Given the high field strength of 7T, the manuscript had better include a discussion on safety consideration, such as specific absorption rate and potential heating effects
The SAR monitoring performed on our 7T MAGNETOM system was based on global power monitoring. However, we have recently shown that the transceiver design even in CP mode achieves local SAR that is within the FDA requirements; this is described in the Methods.
Page 2, Line 95 - 100
The Specific Absorption Rate (SAR) calculation for the transceiver coil was performed in global transmission mode. It should be noted however that even in circularly polarized (CP) mode, the transceiver gives a peak 10-g average SAR of 2.32 W/kg per 1 W of total input power (18), which is within 10 W/kg local SAR for six minutes duration of International Electrotechnical Commission (IEC) guideline (19).
Page 13 - 14, Line 506 - 509
- Kim J, Sun C, Moon CH, Hetherington H, Pan J. Evaluation of the performance of a 7-T 8 × 2 transceiver array. NMR Biomed 2024;37(8):e5146. doi: 10.1002/nbm.5146
- IEC60601-2-33. Medical Electrical Equipment - Particular Requirements for the Basic Safety and Essential Performance of Magnetic Resonance Equipment for Medical Diagnosis. 2022.
R2.4. The authors had better include objective metrics, such as contrast-to-noise ratio, spectral fidelity measures, essential in the evaluating the quality of images produced by their method.
With multi-coil systems, spatial background noise measurements in the image are known to be inaccurate because of each coil has a different spatial sensitivity. Furthermore, a problem in measuring spatial SNR with MRSI (that is distinct from structural or physiological imaging) is that of sufficient outer volume suppression of extracerebral lipids where the noise can be measured. Thus, the best objective metrics for evaluating spectroscopic imaging data remain the reported comparison of metabolite signal, spectral SNR, and CRLB values between the non-Hadamard encoded vs. Hadamard encoded acquisitions. We have restated this point in the Results, reporting the spectral SNR.
Page 6, Line 261 - 266
As the best metric of comparison between the Hadamard-encoded vs. conventional encoded CSI, data from two subjects are shown in a plot of voxel-to-voxel integrated areas of NAA and Cr, demonstrating an R2 = 0.92 (Fig. 2D). Using these inclusion criteria, (n=140 voxels), the SNR of the 7mm-slice thick studies for NAA was 56.6±30.6 vs. 63.9±28.4, and the NAA CRLB was 2.2±0.8 vs. 1.8±0.8 for the CSI and RSI acquisitions respectively which were not significantly different.
R2.5.The author had better conduct a comprehensive error analysis.
To evaluate the accuracy of the rosette spectroscopic imaging (RSI) and the Hadamard encoding, we have compared it to the conventional CSI (standard for this study) using equivalent resolution and acquisition duration. The voxel-by-voxel comparison of the RSI and CSI found a correlation coefficient R2 = 0.92 (Figure 2D). Thus there is excellent agreement between the RSI and CSI. Furthermore, our group has previously published on the accuracy of the fast RSI for the Cramer Rao lower bound (CRLB), evaluating the CRLB with regards to SNR and linewidth (Schirda 2018). As a measure of the coefficient of variation (SD/mean), over the spectroscopic image, the achieved CRLB was found to influence the standard error of regression in relating tissue gray matter with values of Cr/NAA and Ch/NAA. In other words, the ability to detect an abnormal spectroscopic voxel depends on the achieved CRLB, and shows the utility of the CRLB as a measure of spectral error for practical use in patients. Thus we have continued to use the CRLB as a gauge for error and comparison in these Hadamard studies.
Page 6, Line 264 - 266
Using these inclusion criteria, (n=140 voxels), the SNR of the 7mm-slice thick studies for NAA was 56.6±30.6 vs. 63.9±28.4, NAA CRLB was 2.2±0.8 vs. 1.8±0.8 for the CSI and RSI acquisitions respectively which were not significantly different.
Page 7, Line 290 - 296
Despite the significant difference in acquisition time, LCM curve fitting for these spectra shows no significant difference between the CRLB for Glu from either acquisition, with 4.8±1.7% vs. 5.8±2.1% for RSI and CSI, respectively (n=2 subjects, ~130 pixels). For the myo-Ins CRLB, the RSI gave 6.3±3.0%, CSI at 7.0±3.6% and there was no difference in number of pixels meeting inclusion criteria. As expected, with similar filtering criteria as used above (including Glu and myo-Ins CRLB <20%), a significant Glu/NAA regression with fGM is identified in all subjects (Fig. 3D, Table 1).
Round 2
Reviewer 1 Report
Comments and Suggestions for Authors
The authors have improved their manuscript and I recommend it for further processing.